# LRP10, PGK1 and RPLP0: Best Reference Genes in Periprostatic Adipose Tissue under Obesity and Prostate Cancer Conditions

**DOI:** 10.3390/ijms242015140

**Published:** 2023-10-13

**Authors:** Jesús M. Pérez-Gómez, Francisco Porcel-Pastrana, Marina De La Luz-Borrero, Antonio J. Montero-Hidalgo, Enrique Gómez-Gómez, Aura D. Herrera-Martínez, Rocío Guzmán-Ruiz, María M. Malagón, Manuel D. Gahete, Raúl M. Luque

**Affiliations:** 1Maimonides Biomedical Research Institute of Cordoba (IMIBIC), 14004 Cordoba, Spain; b42pegoj@uco.es (J.M.P.-G.); z12popaf@uco.es (F.P.-P.); marina8noviembre@gmail.com (M.D.L.L.-B.); b42mohia@uco.es (A.J.M.-H.); enriquegomezgomez@yahoo.es (E.G.-G.); auritadhm@gmail.com (A.D.H.-M.); bc2gurur@uco.es (R.G.-R.); bc1mapom@uco.es (M.M.M.); bc2gaorm@uco.es (M.D.G.); 2Department of Cell Biology, Physiology, and Immunology, University of Cordoba, 14004 Cordoba, Spain; 3Reina Sofia University Hospital (HURS), 14004 Cordoba, Spain; 4CIBER Physiopathology of Obesity and Nutrition (CIBERobn), 14004 Cordoba, Spain; 5Urology Service, Reina Sofia University Hospital, 14004 Cordoba, Spain; 6Endocrinology and Nutrition Service, Reina Sofia University Hospital, 14004 Cordoba, Spain

**Keywords:** periprostatic adipose tissue (PPAT), gene expression analysis, reference genes, RT-qPCR, prostate cancer (PCa), weight-related disorders

## Abstract

Obesity (OB) is a metabolic disorder characterized by adipose tissue dysfunction that has emerged as a health problem of epidemic proportions in recent decades. OB is associated with multiple comorbidities, including some types of cancers. Specifically, prostate cancer (PCa) has been postulated as one of the tumors that could have a causal relationship with OB. Particularly, a specialized adipose tissue (AT) depot known as periprostatic adipose tissue (PPAT) has gained increasing attention over the last few years as it could be a key player in the pathophysiological interaction between PCa and OB. However, to date, no studies have defined the most appropriate internal reference genes (IRGs) to be used in gene expression studies in this AT depot. In this work, two independent cohorts of PPAT samples (*n* = 20/*n* = 48) were used to assess the validity of a battery of 15 literature-selected IRGs using two widely used techniques (reverse transcription quantitative PCR [RT-qPCR] and microfluidic-based qPCR array). For this purpose, ΔCt method, GeNorm (v3.5), BestKeeper (v1.0), NormFinder (v.20.0), and RefFinder software were employed to assess the overall trends of our analyses. *LRP10*, *PGK1*, and *RPLP0* were identified as the best IRGs to be used for gene expression studies in human PPATs, specifically when considering PCa and OB conditions.

## 1. Introduction

Adipose tissue (AT) is a complex and dynamic organ mainly composed of adipocytes and the stromal vascular fraction, including fibroblast, endothelial cells, preadipocytes, adipocyte-derived stem cells, and a variable infiltrated immune cell population. AT has been classified into white adipose tissue (WAT) and brown adipose tissue (BAT) [1]. While BAT is related to non-shivering thermogenesis through the regulation of lipidic β-oxidation, WAT has been typically conceived as an organ specialized in energy storage. However, rather than acting as a simple lipidic depot, WAT has been recognized as an important contributor to endocrine homeostasis [2]. Through the secretion of the termed adipokines (i.e., any signaling molecule secreted by AT), fat depots are capable of controlling metabolism, inflammation, immune function, and energy homeostasis [3]. For those reasons, studies focused on AT have quickly gained interest, pinpointing it as an active contributor to the onset and progression of several types of endocrine/metabolic disorders such as obesity (OB).

OB is defined as a pathological metabolic condition characterized by an abnormal and excessive lipid accumulation and whose prevalence has drastically increased during the last decades, reaching pandemic proportions. OB is characterized by AT dysfunction and a systemic chronic low-grade inflammatory state which increases the risk for several associated disorders, including insulin resistance, type 2 diabetes, atherosclerosis, cardiovascular diseases, hypertension, and importantly, several endocrine-related cancer types [4]. In fact, according to the International Agency for Research on Cancer (IARC), up to 13 cancer types have been related to overweight/OB conditions, including breast, uterus, colon, kidney, gallbladder, pancreas, rectum, and prostate cancer (PCa), the most studied [5,6]. Regarding PCa, in addition to the mentioned increased risk, OB (along with clinical alterations typically associated with metabolic syndrome such as insulin resistance, elevated blood pressure, and high triglyceride plasma levels) has been linked to a more aggressive disease [7,8,9]. Despite that, the molecular bases linking OB and PCa have not been fully elucidated to date but the number of studies focusing on this pathological association has increased considerably over the last few years.

In this scenario, a specific depot of WAT that surrounds two-thirds of the prostate, typically known as periprostatic adipose tissue (PPAT), has been postulated as a putative active player in the interaction between OB and PCa, not only being a source of bioactive molecules, such as lipids, adipokines, and growth factors (that contribute to microenvironment modulation) but also acting as the first tissue susceptible to be invaded by tumor cells in the early stages of metastasis [10,11]. However, information about the molecular bases underlying that interaction is scarce. In this regard, to explore such a relationship, it is crucial to perform gene expression analyses in PPAT samples derived from patients with different metabolic conditions (e.g., normo-weight [NW] and OB). 

Specifically, reverse transcription quantitative PCR (RT-qPCR) is the gold standard technique for quantifying mRNA expression due to its high accuracy and sensitivity even using small amounts of samples. However, the identification and validation of reliable internal reference genes (IRGs) for normalization of RT-qPCR data are critically needed to correct bias from multiple variables (such as RNA extraction yielding, retrotranscription performance, or primers efficiency, among others) before conclusions can be drawn. In fact, the choice of appropriate IRGs must be confirmed in a tissue-dependent manner and under the specific experimental conditions used in the study. Noteworthily, several studies using multiple rodent models have proven that the choice of IRGs in different AT subtypes depends on the metabolic status [12,13,14]. Regarding humans, although various studies evaluating the use of IRGs in different AT depots have been published [15,16], to the best of our knowledge, there are no reports regarding the identification, evaluation, selection, and validation of suitable IRGs for studying PPAT in expression analyses. Therefore, the objective of this work was to identify and validate, for the first time, reliable IRGs for the normalization of gene expression analyses in PPAT samples under health and disease (PCa and OB) conditions.

## 2. Results

### 2.1. Evaluation of Gene Expression Stability in Periprostatic Adipose Tissue (PPAT) of Patients with Benign Prostate Hyperplasia (BPH) and Prostate Cancer (PCa)

The evaluation of the potential stability in the expression levels of the 15 selected internal reference genes (IRGs) in two independent cohorts of periprostatic adipose tissue (PPAT) samples [i.e., analysis of all samples together from cohort 1 and all samples from cohort 2; Table 1] revealed that the top five IRGs with higher stability (based on changes in the Standard Deviation [SD]) using the RT-qPCR method were *LRP10* (27.99 ± 1.47), *B2M* (23.11 ± 1.50), *RPLP0* (25.97 ± 1.71), *PGK1* (26.64 ± 1.76), and *KDM2B* (31.66 ± 1.78) (Figure 1a; expression levels expressed as SD of Cycle threshold [Ct] values). Statistics details for all IRGs are summarized in Appendix A. Similarly, the top five IRGs with the higher stability using the microfluidic-based qPCR array method were *RPS13* (19.21 ± 1.91), *PGK1* (20.19 ± 1.99), *RPLP0* (18.42 ± 2.01), *IPO8* (25.01 ± 2.02), and *LRP10* (19.84 ± 2.17) [Figure 1b (expression levels expressed as SD of Ct values) and Appendix A)]. It should be mentioned that the use of the SD of the Ct value in these types of analyses/representations is widely used as a first approach since this offers a more visual and direct comparison of the genes by placing them over a narrow dynamic range of values [13,14,17]. In this regard, a representation of the linearized values of the Ct of each of the 15 IRGs (expressed in absolute mRNA copy number) for both techniques used (RT-qPCR and microfluidic-based qPCR array) are also represented in Appendix A, showing similar results. For this latter purpose, standard curves for each selected gene were run in parallel to quantify the absolute mRNA copy number across all the samples analyzed.

On the other hand, the less stable IRGs in terms of variability were *HMBS* (29.02 ± 3.17), *PPIG* (30.17 ± 2.61), *G6PD* (23.89 ± 2.32), *YWHAZ* (26.73 ± 2.21), and *GUSB* (29.54 ± 2.16) using the RT-qPCR method and *HMBS* (23.73 ± 3.46), *PPIG* (24.78 ± 3.03), *GUSB* (22.92 ± 3.01), *HSP90AB1* (20.02 ± 2.91), and *ACTB* (19.71 ± 2.80) using the microfluidic-based qPCR array method (Appendix A).

These results suggest that more stable (less variable) IRG genes using both techniques are *LRP10*, *PGK1*, and *RPLP0* while the worst IRGs (with higher variability) were *HMBS*, *PPIG*, and *GUSB*.

### 2.2. Influence of BMI (Body Mass Index) and PCa Presence on the Expression of IRGs in PPATs

To evaluate the effect of BMI and PCa presence on the stability of the IRG expression levels previously analyzed in PPATs, we stratified the data derived from the two available cohorts of samples under these conditions (Figure 2). Firstly, when BMI was considered (patients with NW vs. OB) we noted that the IRGs previously observed to be more stable (less variable) by both techniques did not exhibit statistically significant differences in their expression levels (i.e., *LRP10*, *PGK1*, and *RPLP0*; Figure 2a), further suggesting that these IRGs are also appropriated candidates to be used in these analyses. In contrast, statistical differences were consistently found in the expression levels of *ACTB*, *B2M*, *GAPDH*, *GUSB*, *HMBS*, and *PPIG* when the samples were classified according to their BMI using both methodologies (RT-qPCR and microfluidic-based qPCR array; Figure 2a), suggesting that these IRGs are not good candidates to be used in these expression analyses.

Secondly, (Figure 2b), when the presence of PCa was considered (patients with BPH vs. PCa), we again observed that the IRGs previously reported to be more stable (less variable) in the different analyses showed until this point (i.e., *LRP10*, *PGK1*, and *RPLP0*; data showed in Figure 1 and Figure 2a) did not exhibit differences in their expression levels (Figure 2a). In contrast, statistical differences were consistently found in the expression levels of *GUSB*, *HMBS*, *IPO8*, *KDM2B*, and *PPIG* (using both RT-qPCR and microfluidic-based qPCR array, methodologies) as well as *ACTB*, *HSP90AB1*, *RPS13*, and *YWHAZ* (using one of the methodologies) when classifying patients in BPH vs. PCa (Figure 2b), suggesting that all these IRGs are not good candidates to be used in these expression analyses.

In order to further explore the influence of the different conditions previously studied [i.e., according to BMI status (NW vs. OB) or presence of PCA (BPH vs. PCa)], different methods specialized in the evaluation of the suitability of IRGs were used [ΔCt, NormFinder, GeNorm, and BestKeeper algorithms]. The results are summarized in Appendix A which shows how the evaluated IRGs are ranked across the two techniques employed by the different software. It should be noted that differences were observed in these rankings depending on the tool used, which was due to the specific algorithms and parameters evaluated in each case. Moreover, it is important to clarify that, in all cases, the three algorithms displayed slightly worse parameters for the data obtained by the microfluidic-based qPCR array compared to the RT-qPCR. This may be due to the fact that, regarding protocol differences, this methodology uses a much smaller amount of starting genetic material; so, a previous pre-amplification step is necessary, which may introduce some kind of bias. Likewise, the pre-amplification step may also explain why the Ct values obtained on the microfluidic-based qPCR array are substantially lower compared to those of RT-qPCR. Nonetheless, in order to solve the discrepancies mentioned, we next used the web tool RefFinder to calculate the geometric mean of ranking values from the weighted values of each algorithm score for each of the IRGs evaluated. Specifically, a comprehensive ranking was generated (Figure 3a) which confirmed previous findings indicating that *LRP10*, *PGK1*, and *RPLP0* are in the top ranking of the most stable IRGs, which reinforces the idea that these three IRGs are the most appropriate to be used in expression analysis of PPATs, at least when considering the presence PCa and OB. Additionally, a bootstrap step strategy without replacement and size 100,000 was performed using the weighted scores previously calculated, confirming that *LRP10* is the best reference gene, followed by *PGK1* and *RPLP0* (Figure 3b).

Importantly, it should be also indicated that a more detailed view of the value distribution of the top three best-ranked IRGs (*LRP10*, *PGK1*, and *RPLP0*) throughout all experimental subgroups of PPATs indicated no statistically significant changes across the different groups (Figure 4). Specific software parameters for all evaluated genes are also listed in Appendix A.

### 2.3. Validation of RPLP0, PGK1, and LRP10 as Reference Genes

Finally, in order to empirically confirm that the selected IRGs were suitable to be used in expression analyses of PPATs obtained under different experimental conditions (i.e., patients with different BMIs or with and without PCa), we measured the expression levels of six metabolic- and tumor-related genes well-known to be dysregulated in the AT of humans under cancer-related conditions (i.e., *SPP1*, *IL6R*, *CD68*, *AGT*, *FNDC5*, and *RETN* [18,19,20,21,22,23]; primers sequences are listed in Appendix A). Specifically, gene expression levels of *SPP1*, *IL6R*, *CD68*, *AGT*, *FNDC5*, and *RETN* were determined using a microfluidic qPCR-based approach and are represented before and after the adjustment by a normalization factor generated by the GeNorm software based on the expression levels of the three selected IRGs (*LRP10*, *PGK1*, and *RPLP0*) (Figure 5). As expected, the expression levels of *SPP1*, *IL6R*, *CD68*, and *AGT* were found to be up-regulated while *FNDC5* and *RETN* levels were down-regulated in PPAT samples from patients with PCa vs. BPH after data normalization. Importantly, these comparisons (non-normalized vs. normalized data) clearly indicated the necessity of using suitable IRGs for the normalization of gene expression levels.

Additionally, to further validate the selection of *LRP10*, *PGK1*, and *RPLP0* as suitable IRGs, we empirically determined the expression levels of two genes (*ANGPT1* and *LEP*) that were previously described to be dysregulated in PPATs of patients with PCa and OB [24]. Specifically, Ribeiro and colleagues indicated that *LEP* and *ANGPT1* were overexpressed in PPATs of patients with BPH under overweight/obese vs. normo-weight conditions, which was confirmed in our analyses using RT-qPCR methodology and normalizing the data with a normalization factor based on the expression levels of *LRP10*, *PGK1*, and *RPLP0*. Likewise, Ribeiro and colleagues demonstrated that the expression of two metalloproteases (*MMP2* and *MMP9*) was increased in conditioned media derived from PPAT explants of overweight PCa patients as compared with normo-weight patients with BPH [25] which was also confirmed in our study after normalization of the data by IRGs (Figure 6c,d). Importantly, these changes observed in the expression levels of *ANGPT1*, *LEP*, *MMP2*, and *MMP9* after normalization were not observed when the data were not normalized (raw data) (Figure 6a–d), which again clearly indicates the necessity of using suitable IRGs for normalization of gene expression levels in PPATs under different experimental conditions (i.e., presence or absence of Ob and PCa).

## 3. Discussion

Gene expression analysis is one of the most used strategies in molecular biology research. However, selecting inappropriate IRGs for the normalization of the data can result in misleading or biased findings. Classically, *GAPDH* or *ACTB* were two of the most commonly used IRGs, but it is well known that these genes can exhibit significant variation in specific cells, tissues, and experimental conditions (such as the metabolic status and/or tumor presence) [15,26], highlighting the need for rigorous selection of IRGs for accurate gene expression analyses and reliable results. 

In this study, we evaluated the suitability of 15 previously documented IRGs in order to determine the most appropriate IRGs for standardizing the mRNA data obtained from PPATs derived from two independent cohorts of samples, using two widely used methodologies (RT-qPCR and microfluidic-based qPCR array) and different algorithms (ΔCt method, GeNorm, BestKeeper, and NormFinder) and also considering the presence or absence of OB and PCa. We believe that this study is relevant because although the relationship between OB and cancer is well known in some specific cancer types, it is still ambiguous in the case of PCa [27] and also because the PPAT has been gaining important attention in recent years as a metabolic- and tumor-driver tissue associated with the prostate gland [28]. However, to the best of our knowledge, no works have evaluated the suitability of different IRGs in PPATs. Moreover, this is also the first study addressing this validation in PPATs from patients with and without PCa and/or OB. 

Firstly, we found that certain IRGs (including *HMBS*, *PPIG*, *GUSB*, and *G6PD*) presented a high variability in their expression levels of PPATs. Indeed, the genes with the greatest variability corresponded to those that presented higher differences in expression levels when the cohorts were distributed according to their BMI and the presence/absence of PCa. Among them, some genes were highly dependent on BMI, such as *ACTB*, *B2M*, and *GAPDH*; some were highly dependent on the PCa existence, such as *GUSB*, *IPO8*, and *KDM2B*; and others varied in both conditions (e.g., *HMBS* and *PPIG*) and were, therefore, considered as the worst IRGs in PPATs under these different experimental conditions.

To evaluate the best IRGs among all the possible candidates, we used four methods based on different statistical parameters: the ΔCt method, GeNorm, BestKeeper, and NormFinder. These analyses revealed some inconsistencies in the results obtained by each software which might be due to different considerations when calculating the scores. For example, the ΔCt method is exclusively based on the average of the SD through the relative comparison of the expression of candidate genes by pairs, GeNorm is based on the arithmetic mean of the pair-wise variation, NormFinder relies on an intra- and inter-group variation of inputted genes, and BestKeeper considers the standard deviation and coefficient variance to calculate the stability factor. Thus, we used a fourth software, called RefFinder, that takes into consideration the weighted contribution score from the previously mentioned software to calculate a geometric mean ranking of values. As a result, we found that the highest stability for RT-qPCR determinations corresponded to the genes *LRP10*, *RPS13*, *PGK1*, *KDM2B*, and *RPLP0* while *PGK1*, *LRP10*, *RPLP0*, *B2M*, and *GAPDH* were the most appropriate for microfluidic-based qPCR arrays; with *LRP10*, *PGK1*, and *RPLP0* being common to both analyses. In fact, we performed a bootstrapping analysis based on the weighted contribution of each software for each gene to their global score which revealed and confirmed that the best genes to be used as IRGs for gene expression studies are *LRP10*, *PGK1*, and *RPLP0*.

Specifically, *LRP10* encodes a low-density lipoprotein receptor family protein but its role in humans is poorly explored. Most available studies associate *LRP10* with neurodegenerative disorders, especially Parkinson’s disease [29]. In agreement with Gabrielsson and colleagues who analyzed a battery of potential IRGs in human AT biopsies in the context of a wide range of commonly studied physiological parameters (including BMI), *LRP10* was found to be the IRG with the highest stability [30]. In the case of *PGK1*, it encodes for a glycolytic enzyme that catalyzes the conversion of 1,3-diphosphoglycerate to 3-phosphoglycerate and exerts an important role in the control of glucose metabolism. Moreover, this enzyme has been widely explored in various fields, including cancer [31,32,33], although it has also been considered a potential IRG in different human tissues under several experimental conditions and even in distinct animal models [34,35]. In this sense, and regarding the adipose tissue depots, Neville and co-workers demonstrated that *PGK1* was the most stable gene among a set of several cohorts of human AT samples and cells, although PPAT was not considered in that work [36]. Lastly, *RPLP0* encodes a ribosomal protein that is a component of the 60S subunit of the ribosomes and has been already proposed to be a suitable IRG across a wide variety of tissues and experimental conditions. Consistent with our results, Zhang et al. evaluated the potential of some genes in three different ATs (epididymal white, inguinal beige, and brown adipose tissue) under different metabolic conditions by using the same software as that used in the present study. Specifically, they showed that the *RPLP0* gene had the lowest variability across all situations evaluated suggesting that could actually be used as a universal IRG in AT [14]. 

In summary, this study confirms the importance of selecting suitable IRGs for normalization to obtain reliable results using different gene expression techniques. It also provides fundamental information regarding a list of potential IRGs that can be used for the analysis of gene expression patterns in PPATs under different experimental conditions. Specifically, this study demonstrates, for the first time, that the expression stability of many IRGs widely used by research laboratories is not valid for the normalization of expression data since their expression significantly varies in PPATs among different experimental settings. Importantly, we solidly demonstrated that *LRP10* has the lowest variability, the highest stability, and more consistency across two widely used techniques (RT-qPCR and microfluidic-based qPCR array); throughout all the analytical software employed, we showed highly stable expression levels in PPAT samples that are independent of the BMI and the presence PCa. In addition, in accordance with the minimum information for publication of quantitative (MIQE) guidelines which recommend the use of multiple reference genes for normalization [37], we provide compelling evidence demonstrating the use of *LRP10*, *PGK1*, and *RPLP0* as the best IRGs for gene expression analyses in human PPATs under different experimental conditions, especially when considering the metabolic status (OB) and the presence of PCa.

## 4. Materials and Methods

### 4.1. Subjects and Samples

This study was approved by the Reina Sofia University Hospital Ethics Committee and conducted following the principles of the Declaration of Helsinki. Cordoba Biobank Node (through Andalusian Biobank) coordinated the collection, processing, and management of the PPAT samples used. Microdisected samples were immediately frozen in liquid nitrogen and then stored at −80 °C following the standard protocols established for this purpose. Written informed consent was obtained from all participants. PPAT samples were obtained from patients who were diagnosed with PCa and immediately treated by radical prostatectomy. All patients were diagnosed with clinically localized and hormone-naive PCa, representing a homogenous cohort of patients (Table 1). 

Two different cohorts were used in this study. The first one (Cohort 1, *n* = 20) was used as a discovery cohort for qPCR determinations. A second larger and independent cohort (Cohort 2, *n* = 48) was used to implement the microfluidic-based qPCR array. Available clinical history details from all patients included in these cohorts are summarized in Table 1, including BMI, Prostate-Specific Antigen (PSA), Gleason score, age, and PI-RADS (prostate imaging reporting and data system). Briefly, four different experimental conditions were analyzed: patients with BPH or PCa and both NW and OB conditions. PI-RADS defines the standards of high-quality for multi-parametric magnetic resonance imaging (mpMRI), including image creation and reporting [38]. Briefly, this system is stratified into five categories which are PIRADS 1, very low (clinically significant cancer is highly unlikely to be present); PIRADS 2, low (clinically significant cancer is unlikely to be present); PIRADS 3, intermediate (the presence of clinically significant cancer is equivocal); PIRADS 4, high (clinically significant cancer is likely to be present); and PIRADS 5, very high (clinically significant cancer is highly likely to be present).

### 4.2. Total RNA Isolation and cDNA Synthesis

Total RNA was isolated with TRIzol Reagent (ThermoFisher Scientific, Bedford, MA, USA) from 100 mg of PPAT following supplier instructions. After tissue disruption, an additional step consisting of two consecutively centrifugations (5 min at 12,200× *g*-force) and supernatant discarding was performed to remove cellular debris and fat excess. A DNAse treatment (RNase-Free DNase Kit—Qiagen, Madrid, Spain) was applied previously to RNA concentration and purity determination on a Nanodrop One Microvolume UV–Vis Spectrophotometer (ThermoFisher Scientific, Bedford, MA, USA). A total amount of 1 µg per sample was then retrotranscribed using random hexamer primers and the RevertAid RT reverse transcription kit (ThermoFisher Scientific, Bedford, MA, USA). Retrotranscribed cDNA was then stored at −20 °C until use.

### 4.3. Potential Reference Gene Selection and Primer Design

For the final selection of the genes included in this study, a combinatorial integration of different bibliographic approaches was carried out. First, an intensive literature search was performed in PubMed, SciELO, and Medline databases using the terms “reference gene” and “housekeeping gene”. Second, we also screened by the terms “adipose tissue”, “cancer”, and “diet” and evaluated two types of articles: (i) those articles whose main objective was to specifically assess the validity of reference genes in human adipose tissues under different cancer and/or diet conditions and (ii) general articles wherein different reference genes were commonly and routinely used in adipose tissues. Finally, and due to the clear limitation in finding information/references regarding human PPATs, we also checked the available information/references of related adipose tissues from rodent models (mouse and rat), including inguinal or perigonadal adipose tissues. Based on all these bibliographic searches, as well as preliminary results from our group using PPATs (unpublished), a list of 15 candidate genes was selected to be studied as potential IRGs in PPATs. The IRGs consulted in this bibliographic approach are included in Appendix A while the 15 candidate IRGs selected, primer sequences, amplicon size, and gene functions are listed in Table 2. Primers that specifically target these genes were designed with the Primer3 software v.0.4.0 while the validation for the suitability for RT-qPCR and microfluidic-based qPCR array technology was assessed by checking the specificity (agarose gel band size, sanger sequencing, and melting curves) as previously reported by our laboratory [39,40]. Specific custom oligonucleotide sequences were purchased from Metabion (Munich, Germany). 

### 4.4. RT-qPCR and Microfluidic-Based qPCR Array

Quantitative RT-qPCR was used to determine the mRNA levels of the selected IRGs in the discovery cohort 1 of PPATs. Specifically, primers for the selected IRGs were used in combination with Brilliant III Ultra-Fast SYBR Green (ThermoFisher Scientific, Bedford, MA, USA) using the Stratagene Mx3000P system (Agilent Technologies, Madrid, Spain) following the manufacturer’s instructions. mRNA levels were determined to analyze the putative steady levels or the variability in the expression across the different groups of patients [i.e., with BPH or PCa and both under NW and OB conditions] using the MxPro 3.0 qPCR software (Agilent Technologies, Madrid, Spain).

Likewise, a microfluidic-based qPCR array (Standard BioTools, San Francisco, CA, USA) was used to simultaneously determine the gene expression levels of the 15 candidate IRGs in the second cohort of PPAT samples using a 48.48 Dynamic Array Plate (GE 48.48 Dynamic Array Reagent Kit with Control Line Fluid, Standard BioTools, San Francisco, CA, USA). Preamplification, exonuclease I treatment, and microfluidic-based qPCR array were implemented as previously described [41,42] following the manufacturer’s instructions using the Biomark HD System and the Fluidigm Real-Time PCR Analysis v4.8.2 Software.

### 4.5. Evaluation of Putative IRGs

To study the suitability of the selected IRGs to be used for the normalization of gene expression analyses as well as to establish a prioritized ranking of these IRGs, a combination of different software analysis (including ΔCt method, NormFinder v.20.0, GeNorm v3.5, and BestKeeper v1.0) were used. Specifically, the ΔCt method is based on the relative comparison expression of ‘pairs of genes’ within each sample. On this basis, stability of the candidate housekeeping genes was ranked according to repeatability of the gene expression differences [43]. The NormFinder algorithm [44] relies on intra- and inter-group variation to estimate gene expression stability enabling the ranking of the candidates so that the less stable genes represent the worse as reference genes. GeNorm [45] calculates the called ‘M’ value using the arithmetic mean of the pair-wise variations of each candidate gene. A low M value (default value threshold set on 1.5) represents a stable gene expression. In addition, GeNorm also calculates the normalization factor according to the number of reference genes inputted. BestKeeper [46] considers the SD as a measure of estimated variability along with the coefficient of variance (CV) from the direct Ct values of each gene. The Ct is established as the number of cycles needed to reach the fluorescent signal threshold so that the lower the Ct level, the greater amount of nucleic sample in an inversely proportional way. Furthermore, RefFinder (http://blooge.cn/RefFinder/), (accessed on 2 September 2023) a web-based tool, was also used to assign a weighted and individual gene score by calculating the geometric mean of their weights for the overall final ranking, allowing the integration of the three computational programs cited below [47]. In addition, a bootstrap step with replacement [48] was performed to evaluate the confidence of the ranking (size of 100,000) across those weighted scores by using the R (v. 3.5.2) package Boot (v. 1.3-28.1). 

### 4.6. Statistical Analyses

Statistical differences were assessed by a *t*-test or by two-way ANOVA followed by Fisher’s correction exact test according to the comparison. All statistical analyses and plots were generated using Prism software 9.0 (GraphPad Software, La Jolla, CA, USA). The *p*-value of < 0.05 was considered statistically significant. Data represent the median (interquartile range) or means ± SD, wherein asterisks (* *p* <0.05; ** *p* < 0.01; and *** *p* < 0.001) indicate statistically significant differences between different conditions. The Metaboanalyst 5.0 online platform was used to perform normalization and density plot visualization.

### 4.7. Study Limitations

Limitations of this study include the selection of a defined set of genes mainly based on a literature search. Additionally, the number of samples (*n* = 20 in the discovery cohort-1 and *n* = 48 in the validation cohort-2) might be considered limited. However, it should be mentioned that the work herein presented represents a collection of PPAT tissues over a 5-year period where the availability of PPAT is dependent on the written informed consent from the participants which, in our experience, is not easy to obtain in many cases. Therefore, it is very difficult to obtain human PPAT samples for research purposes and this is probably the reason why the publications that include PPAT samples are very limited and include similar or lower numbers of samples than the one used in our study [49,50,51]. Furthermore, across the selected patients, other confounding factors such as androgen dependence, tumor behavior, disease duration, or previous anti-tumor treatments, could not be considered in this study. Finally, regarding the validation of the three genes selected as the most suitable IRGs among all the candidates analyzed, part of this validation is based on gene alterations that were described in other types of adipose tissues. Given the peculiar nature of periprostatic adipose tissue, the changes observed may not necessarily occur in the same way as in other depots. Therefore, results herein obtained should be corroborated in additional cohorts of samples.

## Figures and Tables

**Figure 1 ijms-24-15140-f001:**
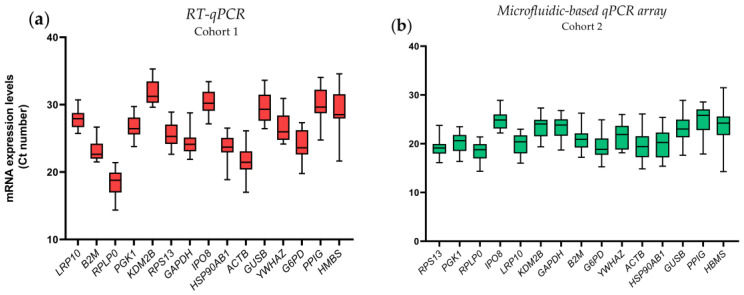
mRNA levels of 15 evaluated internal reference genes (IRGs) in human PPATs. Data (mean ± SD of Cycle threshold (Ct) levels) obtained by the reverse transcription quantitative PCR (RT-qPCR technique) (**a**) or by the microfluidic-based qPCR array technique (**b**). Genes are ordered ascendingly according to the standard deviation.

**Figure 2 ijms-24-15140-f002:**
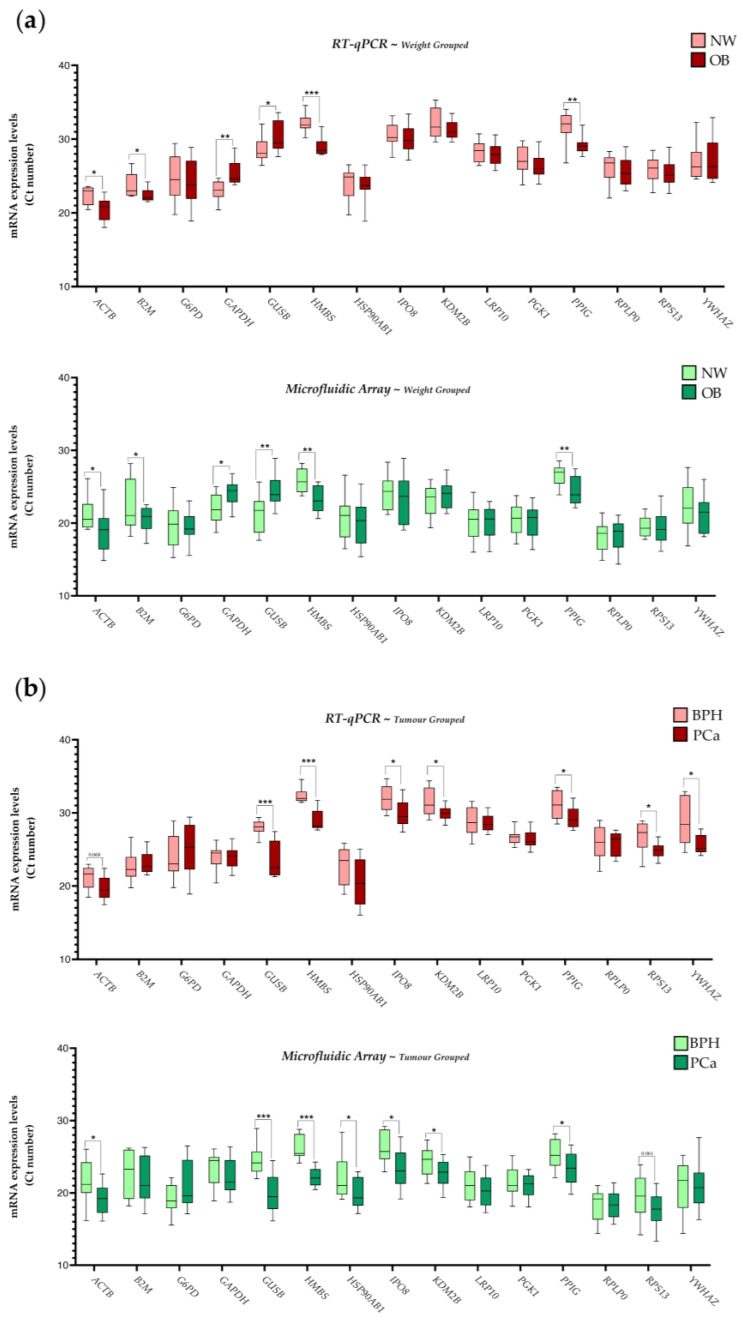
mRNA levels of the 15 evaluated IRGs in human PPAT samples from the two available cohorts of samples stratified by BMI or by the presence of PCa. Data obtained by the RT-qPCR technique (**a**) or by the microfluidic-based qPCR array technique (**b**). Asterisks (* *p* < 0.05; ** *p* < 0.01, and *** *p* < 0.001) indicate statistically significant differences between groups. Normo-weight—NW; Obesity—OB.

**Figure 3 ijms-24-15140-f003:**
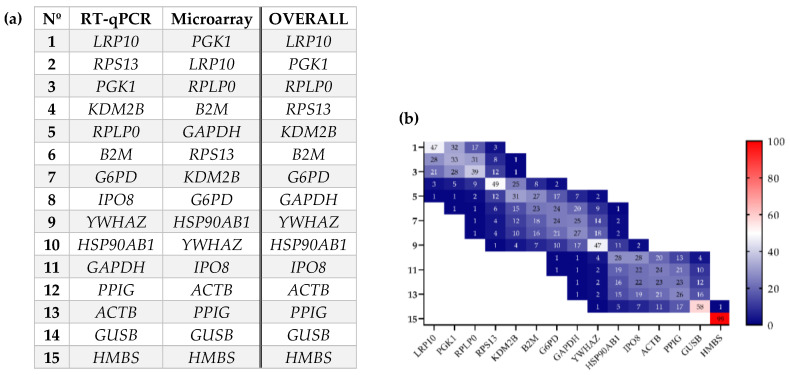
Ranking of the most stable IRGs in human periprostatic adipose tissues (PPAT) derived from the two cohorts of patients BPH and PCa with NW or OB. (**a**) Weighted gene ranking suitability of IRGs evaluated across the ΔCt method, GeNorm (v3.5), BestKeeper (v1.0), and NormFinder (v.20.0) software. (**b**) Empirical distribution of ranks shown in percentage (%) for the 15 evaluated IRGs in bootstrap.

**Figure 4 ijms-24-15140-f004:**
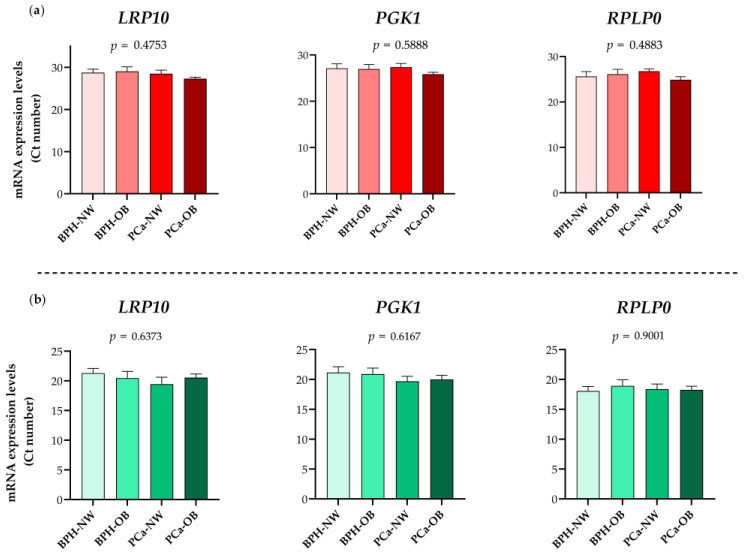
Ct value distribution of the top 3 best-ranked internal reference genes (IRGs) according to Body Mass Index (BMI) or by the presence of prostate cancer (PCa). Expression levels (Cts) were obtained by the RT-qPCR technique (**a**) or by the microfluidic-based qPCR array technique (**b**). Normo-weight—NW; Obesity—OB; Benign Prostate Hyperplasia (BPH).

**Figure 5 ijms-24-15140-f005:**
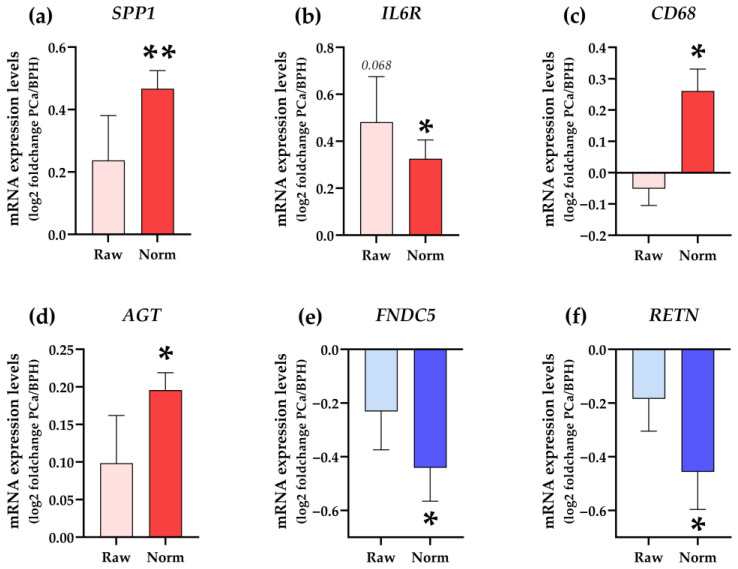
Changes in the expression levels of (**a**) *SPP1*, (**b**) *IL6R*, (**c**) *CD68*, (**d**) *AGT*, (**e**) *FNDC5*, and (**f**) *RETN* in human PPAT from patients with PCa compared with patients with BPH before normalization (raw data) and after normalization (norm) with selected IRGs (*LRP10*, *PGK1*, and *RPLP0*) are shown. Normalization of genes was performed using a normalization factor obtained with the GeNorm software (v3.5) based on the expression levels of *LRP10*, *PGK1*, and *RPLP0*. Asterisks (* *p* < 0.05, and ** *p* < 0.01) indicate statistically significant differences between groups.

**Figure 6 ijms-24-15140-f006:**
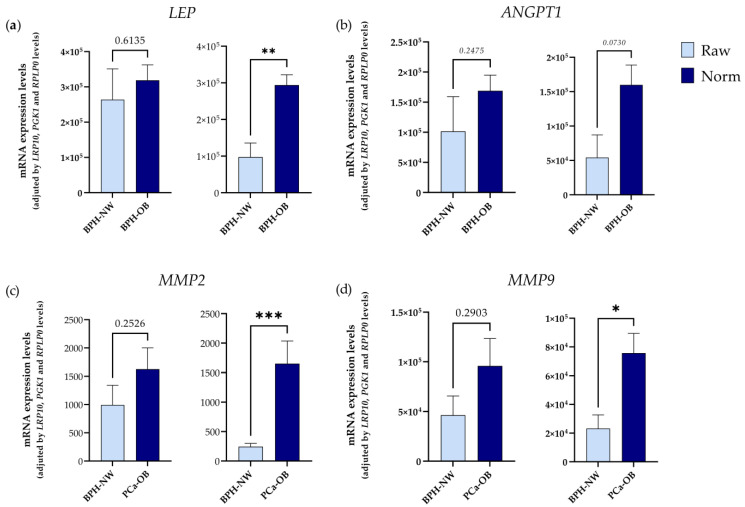
Empirical confirmation demonstrates that the selected IRGs (*LRP10*, *PGK1*, and *RPLP0*) were suitable to be used in expression analyses in human PPAT obtained under different experimental conditions. Expression levels of (**a**) *LEP*, (**b**) *ANGPT1*, (**c**) *MMP2*, and (**d**) *MMP9* before normalization (raw data) and after normalization with selected IRGs (*LRP10*, *PGK1*, and *RPLP0*) are shown. Normalization of genes was performed using a normalization factor obtained with the GeNorm software (v3.5) based on the expression levels of *LRP10*, *PGK1*, and *RPLP0*. Asterisks (* *p* < 0.05; ** *p* < 0.01, and *** *p* < 0.001) indicate statistically significant differences between groups.

**Table 1 ijms-24-15140-t001:** Periprostatic adipose tissue (PPAT) human cohorts and methodologies employed. BMI—Body Mass Index; BPH—Benign Prostate Hyperplasia; PCa—Prostate Cancer; PSA—Prostate-Specific Antigen; SD—Standard Deviation. Data represent the mean ± SD of each group.

Methodology	BMI	Parameter	BPH Group	PCa Group
RT-qPCR Cohort 1 (*n* = 20)	Normo-weight	n	5	5
		BMI (kg/cm^2^)	23.76 ± 1.71	23.47 ± 1.39
		PSA (ng/mL)	4.28 ± 5.67	6.73 ± 4.28
		Gleason score	-	7 ± 0
		Age	69.9 ± 5.59	60.8 ± 4.96
		PI-RADS	2.67 ± 1.15	4.25 ± 0.50
	Obesity	n	5	5
		BMI (kg/cm^2^)	31.05 ± 2.09	33.63 ± 1.25
		PSA (ng/mL)	9.60 ± 17.04	14.29 ± 8.14
		Gleason score	-	7 ± 0
		Age	69.4 ± 7.66	62.2 ± 3.42
		PI-RADS	2 ± 0	4.20 ± 1.30
Microfluidic-based qPCR array Cohort 2 (*n* = 48)	Normo-weight	n	6	14
		BMI (kg/cm^2^)	24.161 ± 3.20	23.80 ± 1.12
		PSA (ng/mL)	3.99 ± 5.12	8.43 ± 7.35
		Gleason score	-	7 ± 0
		Age	69.33 ± 5.04	59.45 ± 5.15
		PI-RADS	2.33 ± 0.81	4.51 ± 0.53
	Obesity	n	14	14
		BMI (kg/cm^2^)	31.26 ± 2.07	31.13 ± 1.01
		PSA (ng/mL)	6.99 ± 10.31	7 ± 4.95
		Gleason score	-	6.85 ± 0.37
		Age	69.22 ± 6.62	58.85 ± 5.33
		PI-RADS	2 ± 0	4.27 ± 0.90

**Table 2 ijms-24-15140-t002:** Gene symbol, primers sequences, Amplicon Size (A.S.), official name, and function of evaluated genes.

Gene	Sequence	A.S (bp)	Official Name	Function
*ACTB*	ACTCTTCCAGCCTTCCTTCCT	176	Actin Beta	Highly conserved proteins involved in cell motility, structure, and integrity
	CAGTGATCTCCTTCTGCATCCT			
*B2M*	GCTCGCGCTACTCTCTCTTT	88	Beta-2-Microglobulin	MHC class I associated protein that is found on nearly all cellular surfaces
	TCCATTCTCTGCTGGATGAC			
*G6PD*	GCAAACAGAGTGAGCCCTTC	89	Glucose-6-Phosphate Dehydrogenase	Cytosolic enzyme whose main function is to produce NADPH using glucose
	GCCAGCCACATAGGAGTTG			
*GAPDH*	GCCTCAAGATCATCAGCAATG	90	Glyceraldehyde-3-Phosphate Dehydrogenase	Enzyme related to carbohydrate metabolism and DNA glycosylation
	CTTCCACGATACCAAAGTTGT			
*GUSB*	TGACCGCTATGGGATTGT	120	Glucuronidase Beta	Enzyme able to hydrolyze glycosaminoglycans
	CTACGCACCACTTCTTCCA			
*HMBS*	TTGCTATGTCCACCACAGG	117	Hydroxymethylbilane Synthase	Enzyme involved in the condensation of porphobilinogen
	CCAGGTCCACTTCATTCTTCT			
*HSP90AB1*	ATGGAAGAGAGCAAGGCAAAG	114	Heat Shock Protein 90α Family Class B Member 1	Constitutive form of the cytosolic 90 kDa heat-shock protein
	GCAGCAAGGTGAAGACACAA			
*IPO8*	CCAAGGGGTGGTTCATTCT	120	Importin 8	GTPase involved in the nuclear import of proteins
	TCTTGCCACAGCTCTTCATC			
*KDM2B*	TGGAGGGCAAAGATTTCAAC	86	Lysine Demethylase 2B	Component of SCFs complex involved in ubiquitination
	TCCCAGTCCATCCTTTTCTC			
*LRP10*	GGCAACGTCACCATCACTT	82	LDL Receptor Related Protein 10	Encodes for a low-density lipoprotein receptor family protein
	AATCTTGGCTGTAGGAGAGCA			
*PGK1*	GGTGGAATGGCTTTTACCTT	83	Phosphoglycerate Kinase 1	Glycolytic enzyme that catalyzes the conversion of 1,3-diphosphoglycerate
	ATCTTGGCTCCCTCTTCATC			
*PPIG*	TTGCGGAGGTACGGATACT	104	Peptidylprolyl Isomerase G	Protein involved in protein peptidyl-prolyl isomerization
	TGAGGAGGAGGAAGATGATG			
*RPLP0*	AACTCTGCATTCTCGCTTCC	119	Ribosomal Protein Lateral Stalk Subunit P0	Ribosomal protein that is a component of the 60S subunit
	GGACTCGTTTGTACCCGTTG			
*RPS13*	CCCACTTGGTTGAAGTTGAC	86	Ribosomal Protein S13	Ribosomal protein that is a component of the 40S subunit
	CCGATCTGTGAAGGAGTAAGG			
*YWHAZ*	TACCGTTACTTGGCTGAGGTT	118	Y3-Monooxygenase/W5-M.oxygenase Act. Prot. Z	Signal transduction mediation by binding to phosphoserine-containing proteins
	GATGTGTTGGTTGCATTTCC			

## Data Availability

The data presented in this study are available in the article and Appendix A.

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
