# Peer review of "LRP10, PGK1 and RPLP0: Best Reference Genes in Periprostatic Adipose Tissue under Obesity and Prostate Cancer Conditions"

_ijms, 2023, doi:10.3390/ijms242015140_

Round 1

Reviewer 1 Report

Authors aimed to identify and validate reliable internal reference genes for normalization of gene expression analyses in periprostatic adipose tissue of human samples. To achieve this aim, authors collected human samples with benign prostate hyperplasia and prostate cancer in normoweight and obese conditions.

I would like to emphasize that manuscript address relevant scientific questions, ensuring scientific rigor and clarity. It is grounded in scientific rational and is structured in a manner that promotes a clear and logical presentation of the research findings.

However, some aspects are highlight below:

-       I suggest to add detailed information about sample characteristics and clinical history, e.g., age, stage of disease, the treatments done, medication, … this suggestion was made based on the high variability that characterize this type of sample.  

-       I do not understand how do you validate the IRGs? The validation of reliable IRGs is too ambitious for the present manuscript.

-       I suggest adding “limitations” at the end of the discussion section.

Author Response

Letter of response to Reviewer’ comments

We sincerely thank the Editors and the Reviewers for the positive and constructive comments on our manuscript, which we found very helpful in improving the quality of our work. Accordingly, based on these comments, specific changes have been made in the manuscript, as described in detail below in a point-by-point description of the changes introduced, and how the Reviewer’s concerns were addressed. Changes within the manuscript are indicated in red.

We honestly trust that the new version of the manuscript after revision, and the responses provided will help to strengthen the support of the Reviewers and hope that this revised version of our manuscript fits within the high-quality scientific standards of the International Journal of Molecular Sciences.

Responses to Reviewer 1

Author’s comment to the reviewer:  We thank the Reviewer for the helpful comments and corrections provided. We have carefully reviewed all these comments and added all the relevant information suggested by the reviewer. The following are responses to the specific comments raised (page and line numbers with the new information added are shown in parentheses to facilitate the tracking of the changes):

* Comment #1: “I suggest adding detailed information about sample characteristics and clinical history, e.g., age, stage of disease, the treatments done, medication, … this suggestion was made based on the high variability that characterizes this type of sample”

Author´s response: Following the reviewer´s suggestion, we have included additional, available, information in Table 1 (page 3). Specifically, we have included the age of the patients, and the PI-RADS (Prostate Imaging Reporting and Data System), which defines the standards of high-quality for multi-parametric Magnetic Resonance Imaging (mpMRI), including image creation and reporting [1]. Briefly, this system is stratified into 5 categories which are: PIRADS 1- Very low (clinically significant cancer is highly unlikely to be present); PIRADS 2 - low (clinically significant cancer is unlikely to be present); PIRADS 3 - Intermediate (the presence of clinically significant cancer is equivocal); PIRADS 4 High (clinically significant cancer is likely to be present); PIRADS 5 - Very high (clinically significant cancer is highly likely to be present). In addition, regarding the likely variability among the patients with prostate cancer (PCa), all of them were diagnosed with clinically localized, and hormone-naive PCa, representing a relatively homogenous selection of patients. This statement has been added in the Materials and Methods section (page 11, 335-338), as follows: “PPAT samples were obtained from patients who were diagnosed with PCa and immediately treated by radical prostatectomy. All patients were diagnosed with clinically localized and hormone-naive PCa, representing a homogenous cohort of patients (Table 1)”. Unfortunately, we have no access to the treatments administered to the patients at the time of sample collection and therefore, this limitation has been included in the newly created “Study Limitations” section (page 14, lines 451 - 453).

* Comment #2: “I do not understand how you validate the IRGs? The validation of reliable IRGs is too ambitious for the present manuscript.”

Author´s response: We appreciate the reviewer´s comment and apologize for not being clear in this regard. The main objective of the work was to solve an experimental problem that had not been previously answered, which is to identify suitable internal reference genes (IRGs) in periprostatic adipose tissue (PPAT) to be used in expression analyses. This identification would be essential to study the pathophysiological association between metabolic disorders (e.g., obesity) and prostate cancer. To this end, we employed two cohorts of samples/patients, a discovery cohort (cohort 1; n=20) and a validation cohort (cohort 2; n=48), wherein the expression levels of 15 different, potential, IRGs were analyzed based on an initial screening of genes widely reported in the literature as valid to be employed as IRGs in multiple human tissues.  Next, different software of analyses specialized in IRGs identification (i.e., GeNorm, BestKeeper, and NormFinder) were used to demonstrate that the expression levels of LRP10, PGK1 and RPLP0 are the most stable to be used as valid IRGs in the PPATs from patients with and without obesity and/or prostate cancer. Finally, in order to demonstrate empirically the validity of these IRGs, we used these three selected IRGs (LRP10, PGK1 and, RPLP0) as normalizing factors of the expression levels of different genes that had been previously reported to be significantly altered in the periprostatic tissues (i.e., LEP, ANGPT1, MMP2, and MMP9), or in other adipose tissues under tumor conditions (i.e., SPP1, IL6R, CD68, AGT, FNDC5, RETN). In this regard, we fully agree that the final section wherein we addressed such validation may be somehow confusing and deserves to be further described. In fact, we strongly believe that this last section is critically important as it allows us to demonstrate that the three selected IRGs (LRP10, PGK1, and RPLP0) are valid as normalizing factors for expression analyses of different genes in PPATs under obesity and/or prostate cancer conditions.

Therefore, and based on the reviewer´s comment, section 2.3 (Validation of RPLP0, PGK1, and LRP10 as reference genes) has been extended/clarified, and additional information has been added to allow a better understanding of this section (pages 7-9). Moreover, new versions of Figures 5 and 6 (pages 8 and 9) have been included comparing the results of the selected genes before and after normalization using the three selected IRGs as normalizers, which clearly demonstrate the necessity of using suitable IRGs for the normalization of gene expression levels in PPATs under different experimental conditions (i.e., presence or absence of obesity and prostate cancer).

* Comment #3: “I suggest adding “limitations” at the end of the discussion section”

Author´s response: We really appreciate this pertinent comment. Following the reviewer´s suggestion, we have included a new paragraph at the end of the manuscript named “Study limitations” (page 14, lines 442 - 458).

REFERENCES

  1. Turkbey, B., A.B. Rosenkrantz, M.A. Haider, A.R. Padhani, G. Villeirs, K.J. Macura, C.M. Tempany, P.L. Choyke, F. Cornud, D.J. Margolis, H.C. Thoeny, S. Verma, J. Barentsz, and J.C. Weinreb, Prostate Imaging Reporting and Data System Version 2.1: 2019 Update of Prostate Imaging Reporting and Data System Version 2. Eur Urol, 2019. 76(3): p. 340-351. doi: 10.1016/j.eururo.2019.02.033

Reviewer 2 Report

The authors conducted a study to verify stable internal reference genes (IRGs) which are those less affected by normal/cancer and normal/obesity variables in human PPAT. In this study, the expression of 15 IRG candidates was measured, and 3 genes were ultimately suggested as stable IRGs. This study appears to have a novelty in that the authors newly suggested stable IRGs in PPAT.

However, it is worrisome that there are some critical aspects hard to agree in this study. To be accepted these concerns should be resolved.

- First, no background was given on what criteria the authors selected 15 IRG candidates to begin the experiment. The criteria and experimental evidence for IRG candidates should be provided. Although, in discussion section, authors indicated that 15 IRG candidates were documented as IRG in somewhere, it is quite unclear what the reasons were. A convincing explanation and references for the basis for selecting IRGs candidates must be provided in the introduction.

- Second, the authors evaluated the stability of IRGs expression by the size of the standard deviation (SD), and the smaller the SD of the Ct value, the more stable the IRGs. Although this interpretation itself can be considered a correct interpretation in general situation, the missed important point is that nobody knows the baseline of the SD value that can be considered stable or not. It is difficult to say whether an arbitrary SD value is large or small without understanding the “SD distribution of the entire gene population”: In analogy, if the SD of the height of a few adults are measured in ‘millimeters’ it would be much larger than that measured in ‘meters’ and it looks very large. So, how to know, for example, if 2.0 of SD of measured Ct values is large or not to decide if it is stable? Even if it is a gene with a relatively large SD value (e.g., HBMS), it is the least stable among the 15 types of genes, and stable/unstable cannot be defined in absolute terms. In the same sense, genes such as LRP10 which are defined as stable IRGs, when judged from the actual entire gene population, there may be genes that are much more stable than these.

- Additionally, Ct value has a non-linear relationship with gene expression level. Therefore, it seems inappropriate to express the variation of Ct values as SD values assuming linearity.

- The number of samples in cohort 1 was 20. Considering the high volatility of human samples, it seems difficult to draw conclusions with statistical significance.

- It is recommended to provide the improvement of using suggested IRGs in normalization. Could the use of three IRGs improved the precision and interpretation of gene expression in PPAT than others? For example, in Figure 5 and 6, the levels need to be compared before and after normalization using IRGs and other candidates to identify if the stable IRGs could correctly normalize the gene expression data and enhance the interpretability of data.

- in page 6, ‘Figure 1’ in legend needs to corrected to Figure 3.

Author Response

Letter of response to Reviewer’ comments

We sincerely thank the Editors and the Reviewers for the positive and constructive comments of our manuscript, which we found very helpful for improving the quality of our work. Accordingly, based on these comments, specific changes have been made in the manuscript, as described in detail below in a point-by-point description of the changes introduced, and how the Reviewer’s concerns were addressed. Changes within the manuscript are indicated in red.

We honestly trust that the new version of the manuscript after revision, and the responses provided will help to strengthen the support of the Reviewers and hope that this revised version of our manuscript fits within the high-quality scientific standards of the International Journal of Molecular Sciences.

Responses to Reviewer 2

Author’s comment to the reviewer:  We really appreciate the comments/suggestions that have been raised by the reviewer about our work, which we believe have served to significantly improve the general quality of the study. The following are responses to the specific comments raised (page and line numbers with the new information added are shown in parentheses to facilitate the tracking of the changes):

* Comment #1: “First, no background was given on what criteria the authors selected 15 IRG candidates to begin the experiment. The criteria and experimental evidence for IRG candidates should be provided. Although, in the discussion section, the authors indicated that 15 IRG candidates were documented as IRG somewhere, it is quite unclear what the reasons were. A convincing explanation and references for the basis for selecting IRGs candidates must be provided in the introduction.”

Author´s response: We really thank the reviewer for this pertinent comment. Following the reviewer´s suggestion, we have included additional information to allow a better understanding of the selection of the internal reference genes (IRGs) used in this study. As indicated in the original version, the selection of the 15 candidate genes was based exclusively on a literature search criterion. In this sense, we would like to clarify that, unlike other widely studied types of tissues, very little information is known about periprostatic adipose tissue (PPAT) since very limited studies are available focusing on PPATs, and to the best of our knowledge, any of them addressing the problem of selecting suitable IRGs to be employed in this specific AT depot. Therefore, this limitation makes it impossible to access other databases or external public cohorts (for example, RNASeqs datasets), which would additionally help to establish different selection criteria when establishing a pre-set of potential candidate genes to study as suitable IRGs. From this perspective, only the literature search was the available option to select this gene pre-set.

Nonetheless, as mentioned above, a more detailed explanation of the selection of IRGs has been added in section 4.3. of the Material and Methods (Page 12, lines 365-380). Briefly, a combinatorial integration of different bibliographic approaches was carried out. First, an intensive literature search was performed in PubMed, SciELO, and Medline databases using the terms "reference gene" and "housekeeping gene". Second, we also screened by the terms "adipose tissue", "cancer", and "diet", and evaluated two types of articles: i) general articles whose main objective was to specifically assess the validity of reference genes in human adipose tissues under different cancer and/or diet conditions; and, ii) general articles wherein different reference genes were commonly and routinely used in adipose tissues. Finally, and due to the clear limitation in finding information/references regarding human periprostatic adipose tissues, we also checked the available information/references of related adipose tissues from rodent models (mouse and rat), including inguinal or perigonadal adipose tissues. Based on all these bibliographic searches, as well as preliminary results from our group using PPATs (unpublished), a list of 15 candidate genes was selected to be studied as potential IRGs in PPATs.  

We have included the key references consulted in this bibliographic approach as a new Supplemental Table 1, while the 15 candidate IRGs selected, primer sequences, amplicon size, and gene functions are listed in Table 2. We hope that these additional clarifications included in section 4.3. of the manuscript, and the new Supplemental Table 1 would better clarify the selection of the IRGs used in this study. In line with this, we have included a new paragraph at the end of the manuscript named “Study limitations” wherein we have included that the selection of the potential IRGs was based exclusively on a literature search (page 14, lines 442 - 443).

* Comment #2: “Second, the authors evaluated the stability of IRGs expression by the size of the standard deviation (SD), and the smaller the SD of the Ct value, the more stable the IRGs. Although this interpretation itself can be considered a correct interpretation in general situation, the missed important point is that nobody knows the baseline of the SD value that can be considered stable or not. It is difficult to say whether an arbitrary SD value is large or small without understanding the “SD distribution of the entire gene population”: In analogy, if the SD of the height of a few adults are measured in ‘millimeters’ it would be much larger than that measured in ‘meters’ and it looks very large. So, how to know, for example, if 2.0 of SD of measured Ct values is large or not to decide if it is stable? Even if it is a gene with a relatively large SD value (e.g., HBMS), it is the least stable among the 15 types of genes, and stable/unstable cannot be defined in absolute terms. In the same sense, genes such as LRP10 which are defined as stable IRGs, when judged from the actual entire gene population, there may be genes that are much more stable than these.”

Author´s response: We appreciate the reviewer´s comment. As the reviewers indicated, the evaluation of the stability of IRGs expression by the size of the standard deviation (SD) is an initially appropriate method to carry out these types of analyses, although it might have some limitations which have been described in the reviewer´s comment. In fact, it should be mentioned that this method of comparison is widely used to select the most appropriate IRGs in adipose tissues (for instance, references: [1-3]). However, due to the mentioned limitations, we performed in the original study additional methods of comparisons using different bioinformatics algorithms (i.e., GeNorm, BestKeeper, and NormFinder; the most widely used methods in these types of analyses).  Moreover, based on the reviewer´s comment, we have now included a fourth mathematical method for assessing the validity of IRGs based solely on the Ct values (i.e., the comparative ΔCt method). The information/data of this additional comparative ΔCt method have been added in the corresponding sections of the article [Materials and method section (page 13, lines 411-415); new Supplemental Figure S2 (where the 4 methods employed are represented); and in the discussion section (page 9, lines 254, 272, 276-277)]. Importantly, all these different comparisons (SD and 4 different bioinformatics/mathematical methods) clearly revealed that the IRGs with the lowest SD were also the best-ranked IRGs. Moreover, further discussion about the representation of the data is included in the next comment of the reviewer.

* Comment #3: “Additionally, Ct value has a non-linear relationship with gene expression level. Therefore, it seems inappropriate to express the variation of Ct values as SD values assuming linearity.”

Author´s response: We thank again the reviewer for this pertinent comment. Certainly, the problem of linearity, which is tightly associated with the previous discussion (comment 2), must be appropriately clarified. As mentioned above, the use of the Ct value in these types of analyses/representations is widely used as a first approach since this offers a more visual and direct comparison of the genes, by placing them over a narrow dynamic range of values. In fact, such comparisons have been previously published in different studies that are also focused on the appropriate choice of IRGs in adipose tissues [1-3].  In this regard, we have added a new analysis included in Supplemental Figure S1, wherein we have represented the linearized values of the Ct of each of the 15 IRGs (expressed in absolute mRNA copy number) for both techniques used (RT-qPCR, and microfluidic-based qPCR array). For this purpose, standard curves for each selected gene were run in parallel to quantify the absolute mRNA copy number across all the samples analyzed. This information has been added to the manuscript [Results (page 3, lines 97-106), and Supplemental Figure S1].

Furthermore, it is important to mention that Supplemental Figure 1 also addresses the issue of the magnitude and scale of the representation of the data (which has been also raised by the reviewer in the previous comment), since the absolute mRNA copy number levels of each IRG is now represented (Supplemental Figure 1) and can be compared with the original representation of the SD of the Ct values (showed in Figure 1). This comparison reveals no differences between both representations. In this sense, it should be indicated that given the exponential nature of the conversion between Cts and absolute mRNA copy number, Supplemental Figure 1 presents the Y-axis in a logarithmic scale in order to facilitate the visualization of the data.

* Comment #4: “The number of samples in cohort 1 was 20. Considering the high volatility of human samples, it seems difficult to draw conclusions with statistical significance.”

Author´s response: We thank the reviewer for this comment. We might agree that n=20 (cohort-1) could be considered a limited number of samples; however, several aspects should be clarified in this regard:

  • This cohort-1 represents the initial discovery cohort, but after obtaining key information in this set of samples, we used an additional, independent, set of samples included in the validation cohort (cohort-2; n=48). Therefore, our analyses were performed in a total of 68 samples. In this context, this is a widely used approach (i.e., use of a discovery cohort and then a validation cohort) for these types of analyses which has previously been used in our group for the analysis of different aspects related to prostate cancer (for instance, reference: [4]);
  • This cohort does not present a high degree of heterogeneity. Specifically, PPAT samples were obtained from patients who were diagnosed with PCa and immediately treated by radical prostatectomy. In fact, all patients were diagnosed with clinically localized and hormone-naive PCa, representing a homogenous cohort of patients;
  • It should be noted that the work presented herein represents a collection of PPAT tissues over a 5-year period, where the availability of PPAT is dependent on the written informed consent from the participants that, in our experience, is not easy to obtain in many cases. Therefore, it is very difficult to obtain human PPAT samples for research purposes, and this is probably the reason why the publications that include PPAT samples are very limited and include similar or lower numbers of samples than the one used in our study [5-7].

Nonetheless, based on the reviewer's comment, we have included this point as a limitation of the study (page 14, lines 442 – 451).

* Comment #5: “It is recommended to provide the improvement of using suggested IRGs in normalization. Could the use of three IRGs improved the precision and interpretation of gene expression in PPAT than others? For example, in Figure 5 and 6, the levels need to be compared before and after normalization using IRGs and other candidates to identify if the stable IRGs could correctly normalize the gene expression data and enhance the interpretability of data.”

Author´s response: Thank you very much for this highly valuable comment. Following the reviewer suggestion, we have compared the levels of the key genes that have been previously described to be dysregulated in adipose tissues of patients under obesity and/or cancer-related conditions (e.g. prostate cancer) [Figure 5: SPP1, IL6R, CD68, AGT, FNDC5, and RETN; Figure-6: LEP, ANGPT1, MMP2, and MMP9)] before and after normalization using a normalization factor obtained from the expression levels of the 3 selected IRGs. These comparisons clearly indicated the necessity of using suitable IRGs for the normalization of gene expression levels in PPATs under different experimental conditions (i.e., the presence or absence of obesity and prostate cancer). This information has been now included as a new Figure 5 and 6 and in section 2.3. of the results (pages 7-9).

* Comment #6: “In page 6, ‘Figure 1’ in legend needs to be corrected to Figure 3.”

Author´s response: Thank you for this correction. Following the reviewer´s comment, we have revised all Figures/Tables titles and updated them according to the new material added in the new version of the manuscript.

REFERENCES

  1. Gong, H., L. Sun, B. Chen, Y. Han, J. Pang, W. Wu, R. Qi, and T.M. Zhang, Evaluation of candidate reference genes for RT-qPCR studies in three metabolism related tissues of mice after caloric restriction. Sci Rep, 2016. 6: p. 38513. doi: 10.1038/srep38513
  2. Krautgasser, C., M. Mandl, F.M. Hatzmann, P. Waldegger, M. Mattesich, and W. Zwerschke, Reliable reference genes for expression analysis of proliferating and adipogenically differentiating human adipose stromal cells. Cell Mol Biol Lett, 2019. 24: p. 14. doi: 10.1186/s11658-019-0140-6
  3. Zhang, W.X., J. Fan, J. Ma, Y.S. Rao, L. Zhang, and Y.E. Yan, Selection of Suitable Reference Genes for Quantitative Real-Time PCR Normalization in Three Types of Rat Adipose Tissue. Int J Mol Sci, 2016. 17(6). doi: 10.3390/ijms17060968
  4. Herrero-Aguayo, V., P. Sáez-Martínez, J.M. Jiménez-Vacas, M.T. Moreno-Montilla, A.J. Montero-Hidalgo, J.M. Pérez-Gómez, J.L. López-Canovas, F. Porcel-Pastrana, J. Carrasco-Valiente, F.J. Anglada, E. Gómez-Gómez, E.M. Yubero-Serrano, A. Ibañez-Costa, A.D. Herrera-Martínez, A. Sarmento-Cabral, M.D. Gahete, and R.M. Luque, Dysregulation of the miRNome unveils a crosstalk between obesity and prostate cancer: miR-107 asa personalized diagnostic and therapeutic tool. Mol Ther Nucleic Acids, 2022. 27: p. 1164-1178. doi: 10.1016/j.omtn.2022.02.010
  5. Altuna-Coy, A., X. Ruiz-Plazas, S. Sánchez-Martin, H. Ascaso-Til, M. Prados-Saavedra, M. Alves-Santiago, X. Bernal-Escoté, J. Segarra-Tomás, and R.C. M, The lipidomic profile of the tumoral periprostatic adipose tissue reveals alterations in tumor cell's metabolic crosstalk. BMC Med, 2022. 20(1): p. 255. doi: 10.1186/s12916-022-02457-3
  6. Roumiguié, M., D. Estève, C. Manceau, A. Toulet, J. Gilleron, C. Belles, Y. Jia, C. Houël, S. Pericart, S. LeGonidec, P. Valet, M. Cormont, J.F. Tanti, B. Malavaud, A. Bouloumié, D. Milhas, and C. Muller, Periprostatic Adipose Tissue Displays a Chronic Hypoxic State that Limits Its Expandability. Am J Pathol, 2022. 192(6): p. 926-942. doi: 10.1016/j.ajpath.2022.03.008
  7. Zhang, Q., L.J. Sun, Z.G. Yang, G.M. Zhang, and R.C. Huo, Influence of adipocytokines in periprostatic adipose tissue on prostate cancer aggressiveness. Cytokine, 2016. 85: p. 148-56. doi: 10.1016/j.cyto.2016.06.019

Round 2

Reviewer 2 Report

Thank you for faithfully explaining many of the points I pointed out. Within the scope of this study, most of the problems presented were resolved. However, the limitations of this study must be accurately communicated to readers to ensure that there are no errors or difficulties in interpreting the results.